# An international multi-centre study to develop and validate federated learning-based prognostic models for anal cancer

Stelios Theophanous [1,2] ✉, Per-Ivar Lønne [3], Ananya Choudhury [4], Maaike Berbee[4], Charlotte Deijen[5,6], Andre Dekker [4], Matthew Field[7,8,9,10], Maria Antonietta Gambacorta[11], Alexandra Gilbert[1,2], Marianne Grønlie Guren [12,13], Rashmi Jadon[14], Rohit Kochhar[15], Daniel Martin[16,17,18], Ahmed Allam Mohamed[19,20], Rebecca Muirhead [21,22], Oriol Parés [23], Łukasz Raszewski[24], Rajarshi Roy[25], Andrew Scarsbrook [1,2], David Sebag-Montefiore[2], Emiliano Spezi [26], Vassilios Vassiliou[27], Eirik Malinen[28,34], Leonard Wee [4,34], Ane Appelt [1,2,34] & atomCAT Consortium*

Precision oncology relies on access to high-quality data for increasingly smaller patient subgroups. The international atomCAT consortium investigates the potential of federated learning to support this, using anal cancer as a rare cancer exemplar. Here, we show that federated multivariable Cox models trained across 14 centres (1428 patients) and externally validated in two additional centres (277 patients) achieve consistent calibration and discrimination during leave-one-centre-out and external validation (c-indices 0.68-0.79). Lower T stage, absence of nodal involvement, smaller tumour volume, female sex, younger age, and mitomycin- or cisplatin-based chemotherapy are associated with improved overall survival. Lower T stage, smaller tumour volume, and female sex are associated with improved locoregional control, while absence of nodal involvement and smaller tumour volume are associated with better freedom from distant metastases. These findings demonstrate that federated learning enables robust, privacy-preserving prognostic modelling for rare cancers using real-world data, supporting international collaboration without data sharing.

Precision oncology has brought increasing use of molecular profiling and sub-stratification, with more patients being treated with targeted agents guided by tumour genetic profiles. Consequently, many previously deemed 'common cancers' (e.g. lung cancer) can now be considered as a collection of rarer tumours. Therefore, rare cancers are diagnosed in an increasingly large proportion of patients[1], and ultimately, all cancers are becoming rare cancers. This poses a key challenge for clinical trials and research: small patient numbers and wide geographic spread make it difficult to generate evidence by standard methods[2]. Recruitment for large clinical trials is complicated and often impossible[3], but traditional registry data do not have the depth needed for detailed subgrouping and analyses. Consequently, this constitutes a major barrier to research and development in an era of stratified medicine.

There has been growing recognition of the significant role of real-world data (RWD) in addressing the challenges related to rare cancer research. RWD can be broadly defined as 'data relating to patient health or experience or care delivery which is collected outside of

---

A full list of affiliations appears at the end of the paper. *A list of authors and their affiliations appears at the end of the paper.
✉e-mail: stelios.theophanous@nhs.net

**Table 1 | Patient and treatment characteristics of the primary and validation cohorts**

| | Primary cohort (14 centres) | Validation cohort (2 centres) |
|---|---|---|
| Number of patients | 1428 | 277 |
| Treatment period | 2004–2022 | 2008–2023 |
| Biological sex | | |
| Male | 435 (30%) | 114 (41%) |
| Female | 993 (70%) | 163 (59%) |
| Age at the start of radiotherapy (years) | | |
| Mean | 62.4 | 61.0 |
| (SD, range) | (11.3, 29–94) | 12.5 (21–92) |
| Age <50 | 197 (14%) | 52 (19%) |
| Age 50–69 | 835 (58%) | 149 (54%) |
| Age >= 70 | 396 (28%) | 76 (27%) |
| T stage at diagnosis | | |
| T1-2 | 815 (57%) | 187 (68%) |
| T3-4 | 613 (43%) | 90 (32%) |
| N stage at diagnosis | | |
| N0 | 697 (49%) | 155 (56%) |
| N+ | 731 (51%) | 122 (44%) |
| M stage at diagnosis | | |
| M0 | 1390 (97%) | 272 (98%) |
| M1 | 38 (3%) | 5 (2%) |
| Primary tumour GTV (cm3) | | |
| Mean | 70.6 | 39.7 |
| (SD, range) | (157.3, 0.6–974.4) | (46.4, 1.4–314.6) |
| Median | 41.5 | 29.6 |
| Histology | | |
| SCC | 1233 (86%) | 250 (90%) |
| Basaloid SCC | 195 (14%) | 27 (10%) |
| Primary tumour dose (EQD2 $\alpha/\beta = 10$) | | |
| Mean | 53.2 | 55.3 |
| (SD, range) | (4.2, 40.7–66.0) | (3.1, 33.6–63.7) |
| Radiotherapy technique | | |
| 3D-CRT | 117 (8%) | 38 (14%) |
| IMRT / VMAT | 1311 (92%) | 239 (86%) |
| Chemotherapy regimen | | |
| No chemotherapy | 83 (6%) | 52 (19%) |
| MMC and 5FU | 927 (65%) | 194 (70%) |
| MMC and Cap | 365 (26%) | 22 (8%) |
| Cispl and 5FU | 12 ( <1%) | 3 (1%) |
| Cispl and Cap | 3 ( <1%) | 0 (0%) |
| Other | 38 (3%) | 6 (2%) |

Summary statistics for patient and treatment characteristics across the entire training cohort (14 centres) and validation cohort (2 centres). Means and medians are all weighted based on centre cohort size. SD Standard deviation, GTV Gross tumour volume, CI Confidence interval, SCC Squamous cell carcinoma, MMC Mitomycin C, 5FU 5-fluorouracil, Cap Capecitabine, Cispl Cisplatin. N/A Information not available.

clinical trials'[4,5]. Given that rare cancers are often dispersed across different healthcare institutions and geographic regions, collating large, high-quality datasets with sufficient depth is complex when reliant on centralised analysis. Patient privacy and data protection in particular are often the key obstacles in this setting[6]. Federated learning (FL), however, enables collaborative analysis without the need to pool data centrally[7,8]. Through this approach, individual institutions retain control over their data, ensuring patient privacy and compliance with data governance regulations. FL with RWD offers a solution to data scarcity for rare cancers, allowing for better understanding of prognostic factors as well as outcomes in patient sub-groups[9,10].

The international atomCAT consortium (Anal cancer Treatment Outcome Modelling with Computer Aided Theragnostics) was established in 2021 with the aim of demonstrating the value of FL in rare cancers, using anal cancer as the exemplar. Specifically, the consortium set out to identify prognostic factors for anal cancer using FL[11], building on previous proof-of-concept work[12]. Anal carcinoma is rare, comprising approximately 0.3% of all cancer cases[13,14]. However, its incidence is rising substantially, with studies showing a 35% increase in men and a 75% increase in women between 1988 and 2012 across Europe[15]. This trend is largely attributed to increased exposure to high-risk HPV, following demographic and behavioural shifts since the introduction of the oral contraceptive pill in the 1960s. Despite the rising incidence, most centres treat relatively few cases per year and may struggle to identify appropriate evidence-based treatment for patient subgroups, many of whom are under-represented in clinical trials. As an example, if an oncologist were to see an elderly patient with anal cancer in their clinic today, the treatment offered might differ substantially across cancer centres and countries[16].

Current standard treatment for localised anal cancer consists of concurrent chemotherapy and radiotherapy[17–19] delivered with intensity-modulated radiation therapy (IMRT) or volumetric modulated arc therapy (VMAT)[19]. This approach confers favourable outcomes: modern UK and US multi-centre cohorts have reported three-year overall survival and disease-free survival rates of 80–85% and 75%, respectively[20,21]. Historically, most disease relapses have been reported to occur locoregionally, i.e., at the site of the primary disease or in involved lymph nodes, although this pattern may be changing with modern treatment strategies[21]. Previous research has conveyed a clear need for improved treatment stratification with respect to locoregional treatment intensification or deintensification, as well as the use of novel agents for systemic disease management. The design of new treatment stratification strategies depends on robust assessment of risk groups with respect to both locoregional and distant metastatic control[22,23].

In this report, we demonstrate the value offered by international real-world curated data research utilising federated learning through a prospectively planned and pre-registered study of a large international anal cancer patient cohort. We report on the development and validation of federated prognostic models for overall survival (OS), locoregional tumour control (LRC), and freedom from distant metastases (FFDM), based on data from anal cancer patients treated with modern chemoradiotherapy at 16 cancer centres across Europe and Australia.

## Results
A total of 1428 patients, treated from 2004 to 2022 across 14 participating centres, were identified for inclusion in the primary cohort analysis, with an additional 277 patients available for external validation. Table 1 summarises the patient characteristics of the primary and validation cohorts; with data stratified on individual centres available in Supplementary Material: Supplementary Note 5. More than 80% of patients were treated between 2010 and 2022.

The overall cohort had a weighted mean age of 62 at the start of radiotherapy, with 28% of patients older than 70 years. The majority of patients (70%) were female. Primary tumour volume varied considerably across centres, from a mean of 23.9 cm$^3$ in Centre 6 to a mean of 120.5 cm$^3$ in Centre 1. Mean prescribed primary tumour dose (in equivalent dose in 2 Gy per fraction, $\alpha/\beta = 10$ Gy, EQD2$_{\alpha/\beta=10Gy}$) was relatively consistent across centres, ranging from 50.6 Gy to 60.1 Gy. The most common chemotherapy regimen was Mitomycin C (MMC) and 5-fluorouracil (5FU)/capecitabine (Cap), which 91% ($n = 1292$) of patients received. Only 6% ($n = 83$) of patients received no chemotherapy.

**Table 2 | Outcome summary statistics for the primary and validation cohorts**

| | Primary cohort (14 centres) | Validation cohort (2 centres) |
|---|---|---|
| Number of patients | 1428 | 277 |
| **Number of events** | | |
| Deaths | 286 (20%) | 44 (16%) |
| Locoregional failures | 214 (15%) | 32 (12%) |
| Distant metastases | 163 (11%) | 30 (11%) |
| **Overall survival** | | |
| 2-year OS | 88% | 89% |
| 3-year OS | 83% | 87% |
| 5-year OS | 76% | 79% |
| **Locoregional control** | | |
| 2-year LRC | 86% | 90% |
| 3-year LRC | 83% | 89% |
| 5-year LRC | 81% | 84% |
| **Freedom from distant metastases** | | |
| 2-year FFDM | 89% | 89% |
| 3-year FFDM | 87% | 89% |
| 5-year FFDM | 86% | 88% |

Summary of survival statistics, including overall number of events, as well as estimated overall survival (OS), locoregional control (LRC), and freedom from distant metastases (FFDM) at two, three, and five years, for the training cohort (14 centres) and validation cohort (2 centres).

Summary survival statistics for the three outcomes are provided in Table 2, and presented visually for all centres in Fig. 1, with full per centre data in the Supplementary Material: Supplementary Note 5. Weighted mean potential follow-up times across the entire cohort were 48.7 months for OS, 42.0 months for LRC, and 41.8 months for FFDM. At 3 years, the weighted means (range) for OS, LRC, and FFDM were 83% (75–97%), 83% (73–90%), and 87% (57–96%), respectively. OS rates varied more than LRC and FFDM rates across centres.

The results from the OS, LRC, and FFDM global models, trained using all available data from all primary cohort centres, are presented in Table 3. Lower T stage, lack of nodal involvement, female sex, younger age, smaller primary tumour, and use of doublet chemotherapy with MMC or Cisplatin and 5FU or Cap (relative to no chemotherapy) were associated with better OS. Moreover, lower T stage, smaller primary tumour size, and female sex were associated with better LRC. Lastly, lack of nodal involvement and smaller primary tumour size were associated with better FFDM.

The multivariable outcome models allow for individual patient risk prediction using the reference outcome rates summarised in Supplementary Material: Supplementary Note 5. Observed weighted mean three-year outcome rates across centres were markedly different between predicted high- and low-risk groups: 73% vs 90% for OS; 76% vs 91% for LRC, and 83% vs 94% for FFDM. Plots for high- and low-risk groups, indicating the observed weighted mean outcome rates across centres at two, three, and five years are presented in Supplementary Material: Supplementary Note 5. Figure 2 demonstrates the failure risk distribution for locoregional control and distant metastases for individual patients in a single centre cohort (Centre 2, *n* = 210); with the impact of tumour volume on failure risk highlighted.

The global model performance on the primary study cohort was acceptable, with weighted mean c-indices of 0.68, 0.71, 0.69 for OS, LRC, and FFDM, respectively. The performance was stable for the leave-one-centre-out internal validation (0.67, 0.69, 0.66), indicating very little overfitting. This was confirmed by consistent model performance in the external validation cohorts (0.72, 0.75, 0.79). Full details of model performance per centre can be found in Supplementary Material: Supplementary Note 5.

To illustrate the added value of the federated approach, comparator models for all three outcomes were developed using the same specifications, but only data from a single centre (Centre 2, *n* = 210). When evaluated on the external validation cohorts, these single-centre models yielded substantially lower C-indices for all outcomes (OS: 0.60, LRC: 0.70, FFDM: 0.62) compared to the federated models (0.72, 0.75, and 0.79, respectively), indicating reduced generalisability.

None of the secondary models (Supplementary Material: Supplementary Note 5) performed considerably better than the primary cohort models. For OS and LRC, the secondary models including performance status and treatment compliance performed minimally better, but at the cost of having additional parameters. This took the total model parameters beyond the number supported by the available sample size. Using a simple split on staging (high risk: T4 $N_{any}$ or $T_{any}$ N + , vs low risk: T1-3 N0), instead of including T stage and N stage individually, was less useful for LRC (where it became non-significant) but still provided relevant information for OS and FFDM. For distant metastases, removing GTV size from the model resulted in a significant effect of T stage on outcome, which was not observed with GTV included. None of the alternative parametrisations of continuous factors (age, GTV size) performed better than the ones used in the primary models.

Figure 3 presents calibration plots for three-year OS, LRC, and FFDM. Points closer to the reference line indicate stronger calibration and signify that the predicted outcome rates correspond more closely to actual (observed) outcome rates. The results indicate that calibration was better for LRC prediction compared to OS and (to a lesser extent) FFDM. Calibration was generally good in the external validation cohorts (indicated with black circles in Fig. 3).

## Discussion

This international consortium study aimed to demonstrate the use of federated learning on real-world data to develop and validate prediction models for anal cancer outcomes after chemoradiotherapy. For three clinically important outcomes, consistent model performance was demonstrated in independent external validation datasets. Importantly, we identified a robust set of prognostic factors with differential effects on survival, locoregional control and risk of distant metastatic disease. Although some of the prognostic variables identified (e.g., T stage, N stage, GTV) are well supported in previous studies, our findings show that federated learning can reproduce and externally validate these variables. This highlights the capability of federated learning for developing robust, privacy-preserving prognostic models for rare cancers using real-world data.

To our knowledge, the multi-national atomCAT cohort of 1705 patients is one of the largest contemporary international anal cancer cohorts analysed to date. We found that lower T stage was associated with better OS and LRC, while nodal involvement was associated with poorer OS and higher risk of distant metastases. Sex was associated with OS and LRC (but not distant metastases), with females doing considerably better than males for both outcomes. These results confirm findings from previous smaller studies, summarised in our systematic review[24]. We demonstrated the importance of primary tumour size as a prognostic factor for all three outcomes examined. This emphasises the importance of imaging-specific factors on cancer outcomes; data which are not usually available e.g. in centralised cancer registries, although noting the overlap of this factor with T stage. Interestingly, our findings suggest that prescribed radiotherapy dose is not prognostic for any of the three outcomes, which contradicts previous reports of radiotherapy dose-response for tumour control based on literature data[25,26]. This lack of dose-effect relationship could reflect cohort heterogeneity and/or confounding by indication, as higher doses tend to be prescribed to patients with more advanced disease. In addition, variations in dose reporting across institutions may dilute this relationship. Prospective, randomised

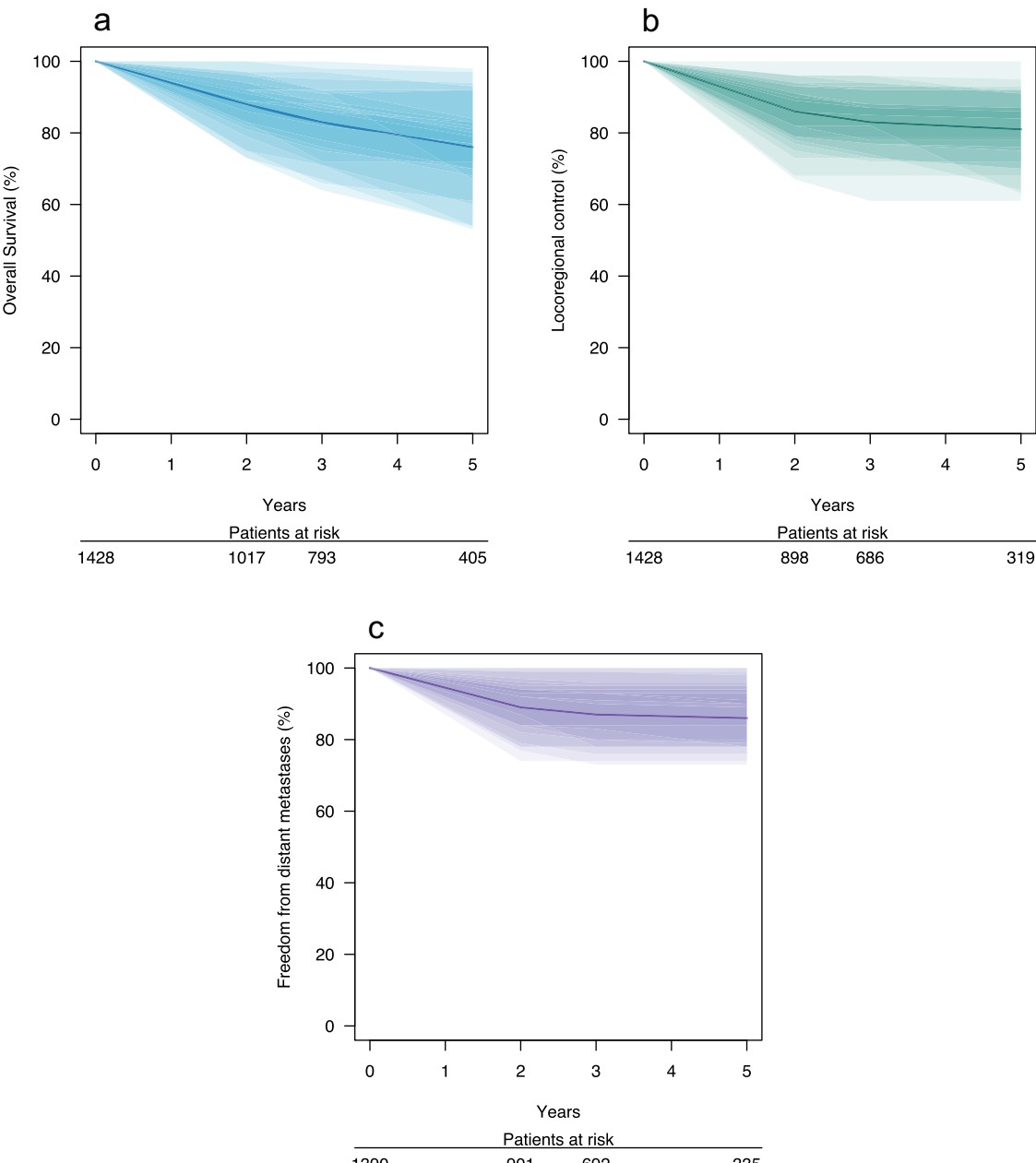

**Fig. 1 | Outcome rates across participating centres. a** Overall survival,
(**b**) locoregional control, and (**c**) freedom from distant metastases rates, calculated using Kaplan-Meier methods. Patients at risk at two years, three years, and five years are specified below. Shaded areas represent the 95% confidence interval bands from 12 participating centres in the primary cohort, while the solid line represents the weighted mean across all 12 centres. Primary cohort centres 6 and 14 were not included in this figure due to lack of 5-year follow-up data. Source data are provided as a Source Data file.

studies with risk-adapted doses such as PLATO[27] are expected to clarify this issue.

While the discrimination metrics from the outcome models were moderate (c-indices 0.68–0.79), these values are comparable with other externally validated real-world prognostic models in oncology in general and anal cancer specifically[28–30]. They indicate clinically meaningful performance in a rare cancer setting, where predictive model performance is often limited by cohort size and heterogeneity. Moreover, slightly poorer calibration for OS and FFDM relative to LRC was observed, which likely reflects greater heterogeneity in non-cancer mortality across centres. Despite this, model performance remained within acceptable limits for clinical interpretation. Formal statistical comparisons between the training and validation cohorts were not performed, as these were independent datasets used to test model

generalisability across different populations. The observed differences in demographics and treatment reflect real-world variability, which the federated model is designed to accommodate.

Our achievement was galvanised by use of RWD connected through FL, avoiding complicated data protection issues and data sharing agreements. Although comparison to a fully centralised model was not feasible due to data governance constraints, the federated Cox regression algorithm used in this study has been shown to be mathematically equivalent to a centralised model under the Breslow approximation when applied to the same data[31]. As such, this study demonstrates the potential of the FL approach for generation of robust evidence for rare cancers, where high-quality data for large patient cohorts are otherwise challenging to access. Given the increasing focus on molecular sub-stratification of many cancers, this

**Table 3 | Multivariable Cox proportional hazards model results**

|  | Overall survival | Locoregional control | Freedom from distant metastases |
|---|---|---|---|
| Mean global model c-index | 0.68 | 0.71 | 0.69 |
| Mean leave-one-centre-out validation c-index | 0.67 | 0.69 | 0.66 |
|  | Hazard ratio (95% CI) |  |  |
| Nodal involvement (N+ relative to N0) | 1.45 (1.11–1.89) | 1.24 (0.92–1.68) | 2.09 (1.42–3.08) |
| T stage (T3-4 relative to T1-2) | 1.42 (1.07–1.89) | 1.46 (1.05–2.03) | 1.18 (0.80–1.74) |
| Biological sex (Female relative to male) | 0.65 (0.51–0.83) | 0.56 (0.43–0.73) | 0.82 (0.58–1.16) |
| Age at start of radiotherapy (per 10 years) | 1.20 (1.07–1.34) | 1.08 (0.96–1.22) | 1.00 (0.86–1.16) |
| Gross tumour volume ($cm^3$) | 2.02 (1.47–2.76) | 2.47 (1.73–3.53) | 2.14 (1.40–3.27) |
| Prescribed dose to primary tumour ($\log_{10}$ EQD2, per 10 Gy) | 0.96 (0.71–1.29) | 1.17 (0.82–1.67) | 1.21 (0.79–1.86) |
| Histology (Basaloid SCC relative to SCC) | 0.88 (0.61–1.28) | 0.64 (0.39–1.06) | 1.04 (0.64–1.69) |
| Radiotherapy technique (IMRT/VMAT relative to 3D-CRT) | 0.96 (0.67-1.39) | 1.55 (0.91–2.64) | N/A |
| Chemotherapy regimen (all relative to no chemotherapy) |  |  |  |
| Mitomycin-based | 0.35 (0.23–0.53) | 0.67 (0.35–1.25) | 0.59 (0.28–1.23) |
| Cisplatin-based | 0.32 (0.11–0.92) | 0.72 (0.22–2.30) | 0.80 (0.21–3.09) |
| Other chemotherapy | 0.81 (0.42–1.56) | 0.83 (0.30–2.27) | 0.94 (0.31–2.92) |

Results from multivariable Cox proportional hazards models for overall survival (OS), locoregional control (LRC), and freedom from distant metastases (FFDM), trained on the full primary cohort (14 centres). Hazard ratios (HRs) and 95% confidence intervals (CI) are shown for each included variable. All models were adjusted for the full set of covariates listed (apart from radiotherapy technique, which was not included in the FFDM model).

will become important for a growing number of patients in the coming years. However, using RWD to explore comparative effectiveness (such as effects of different dose regimens) has numerous potential pitfalls[5], including confounding by indication, and should preferably be considered in a formal causal analysis framework[32]. As an example, the limitations for analysis of radiotherapy dose and its impact on LRC are discussed above. This question has been explored in the PLATO trial[27], which is expected to provide a definitive answer to this question on treatment stratification in the near future. In other words, RWD should be seen as complementary to and not a replacement of clinical trials. It may offer crucial guidance for future trial design, such as identification of particular risk groups, and may augment knowledge from clinical trials[33–35]. Importantly, older patients comprised nearly one-third of our cohort, yet remain poorly studied in prospective trials[16]. Models trained on RWD may therefore offer a tool to support individualised treatment discussions and improve prognostication for such under-represented groups.

In addition to the FL methodology discussed above, the atomCAT consortium study has several other strengths. Firstly, feature selection was based on expert consensus and literature review, rather than automated methods, avoiding known limitations with techniques like LASSO and stepwise selection in smaller, heterogeneous datasets[36–38]. This approach ensured clinical relevance, reproducibility, and supported prospective sample size calculation. The prospective nature of various aspects of this study, including the sample size calculation and study pre-registration, is a particular strength. The former aimed to minimise model overfitting and ensure that the overall risk of each outcome was estimated precisely[39], while the latter guarded against data dredging and post-hoc model tuning[40,41]. Consequently, model performance as well as individual factor effect sizes are likely robust and not over-optimistic. This is supported by consistent model performance in the internal-external ("leave-one-centre-out") and external cohort validation[42]. Additionally, use of a large and diverse set of patients, covering a range of geographical regions and healthcare systems with a variety of standard treatment protocols, increases the likelihood of the findings being generalisable beyond the study population.

Data availability, quality, and curation are well-known challenges for studies of RWD generated in routine care setting, and also underlies some of the main limitations of our study[5]. Some potential prognostic factors were not selected for inclusion in the primary model

analysis due to their unavailability for most patients at many consortium centres. HIV and HPV status were excluded due to inconsistent data availability across centres, and performance status was omitted because of high missingness and potential informative bias. Frailty may ultimately represent a more clinically meaningful predictor of treatment tolerance and outcome than performance status or age; however, its incorporation into federated analyses would require standardised definitions to ensure comparability across centres. The study protocol and data analysis plan adopted a pragmatic approach, exploring factors routinely recorded in daily practice, rather than e.g. emerging or novel biomarkers. This minimised any issues with missing data but may have introduced unknown confounding by indication. Data curation and quality assurance was facilitated by keeping all data within individual centres, where review and validation for individual patient cases were straightforward. This proved to be an additional benefit of the federated learning approach compared to centralised data analysis.

Some of the study limitations are related to technical aspects of the federated learning architecture; especially with respect to standard prognostic model development steps which were not available in our federated implementation at the time of analysis. This includes advanced methods for missing data imputation[43,44] and correlation analysis. Instead, simple imputation methods were used, and correlated variables were handled during the study design phase. For example, secondary models explored the overlap between T stage and GTV size and indicated that some (but not all) of the information from GTV size can be captured by T stage. Due to privacy restrictions within the federated learning framework, individual-level Kaplan-Meier curves stratified by clinical subgroups could not be generated, although future Vantage6 algorithms may allow secure aggregation of subgroup survival estimates. Additionally, although the federated Cox model is mathematically equivalent to the centralised version under the Breslow approximation, small numerical deviations may occur due to asynchronous updates or local preprocessing. These differences are likely to be negligible and do not affect overall model conclusions. Moreover, two centres were excluded from the summary 5-year outcome rates due to incomplete follow-up; however, this represents a small fraction of the total cohort (14 centres), and the two- and three-year endpoints remain unaffected. Subgroup analyses were not pre-specified and were underpowered in the available cohorts. Future federated learning studies with larger datasets will allow more reliable

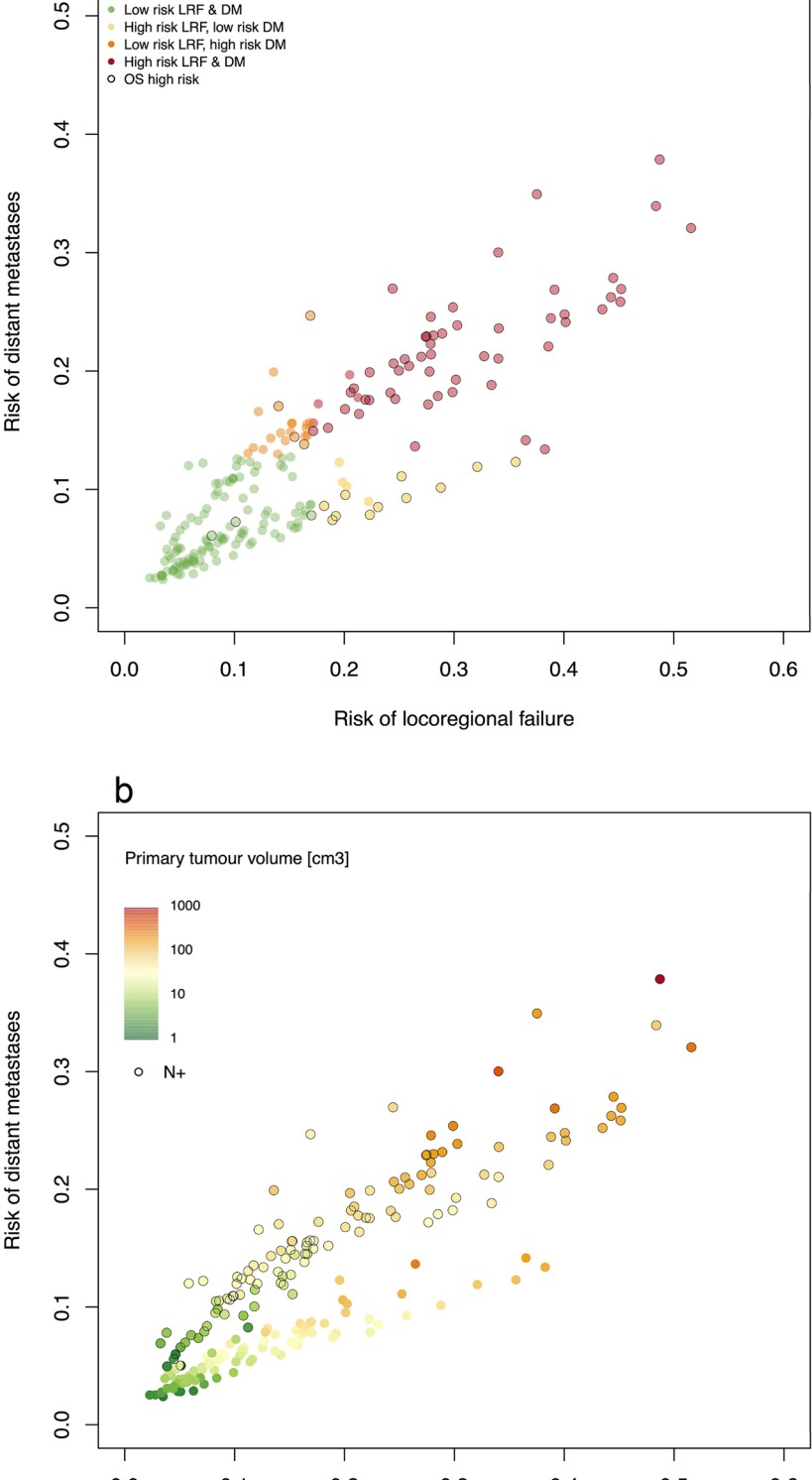

**Fig. 2 | Predicted failure risk distribution and impact of tumour volume.** Failure risk distribution for locoregional control and distant metastases for individual patients in a single centre cohort (Centre 2, $n = 210$). In (**a**), individual patients are stratified into locoregional failure and distant metastasis risk groups according to their predicted risk score; (**b**) highlights the impact of tumour volume on failure risk.

evaluation of subgroups, including older, frailer, and comorbid patients.

Future enhancements to federated learning are likely to come from advances in data curation, interoperability, and infrastructure.

Large language models and multimodal generative pre-trained transformer (GPT) systems could assist with local data cleaning, mapping, and harmonisation, reducing the burden of manual curation. Adoption of common data models (CDMs) and interoperability frameworks,

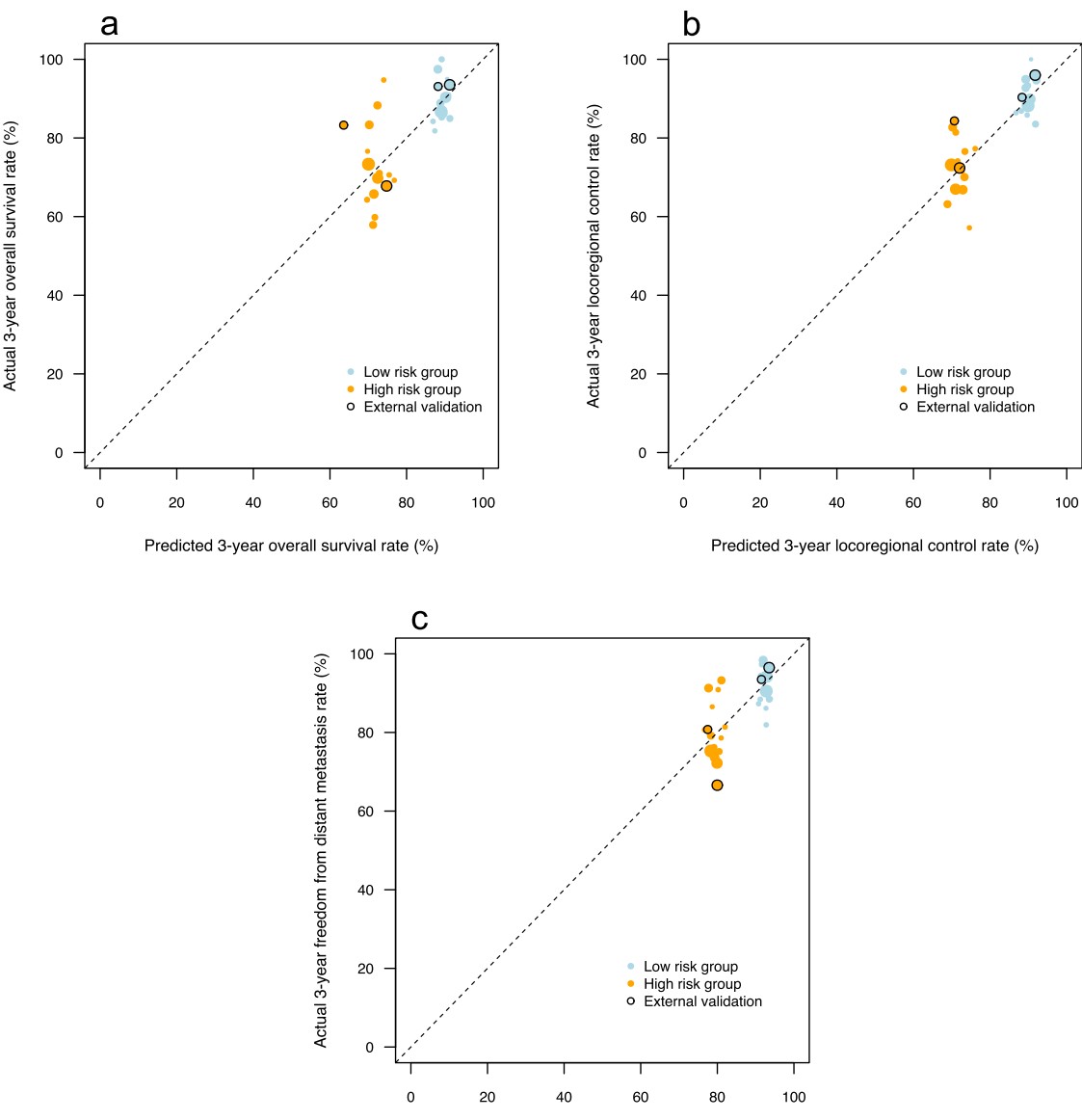

**Fig. 3 | Calibration of federated prognostic models for 3-year outcomes.** Calibration curves for 3-year (**a**) overall survival, (**b**) locoregional control, and (**c**) freedom from distant metastases rates. Size of circles corresponds to individual cohort size. The black circles represent the calibration results from the two external validation cohorts. Source data are provided as a Source Data file.

such as the OMOP CDM[45] and the operational ontology for oncology (O3)[46], can enable collaboration and analysis across heterogeneous clinical datasets. Generative AI could also be deployed locally to create synthetic patient data to support imputation or mitigate inclusion bias without sharing identifiable information. The Vantage6 platform supports containerised implementation of a large range of model types, including machine learning and deep learning algorithms. Recent examples include federated survival forests and convolutional neural networks for radiotherapy and imaging research[47], illustrating the potential for broader applications as the infrastructure matures further.

In conclusion, the atomCAT consortium analysed a large, geographically distributed multi-centre anal cancer cohort. The study provides unique insights into the distinct prognostic effect of different patient and disease characteristics on OS, LRC, and FFDM, and may inform the design of future clinical trials and risk stratification for treatment personalisation. It demonstrates the value offered by high-quality international real-world data research utilising federated learning. This approach may now also be used for more complex research questions and data, such as tissue, blood and imaging biomarkers or additional radiotherapy-specific data. Our robust and validated prognostic models, developed using large cohorts, lay the foundation for decision support tools that can inform shared decision-making, enable more personalised treatment strategies, and may be considered in prospective clinical trials for patient inclusion and stratification. In the future, this federated approach could also guide biomarker-driven subgroup analyses and support policy development, such as reimbursement decisions for new therapies and informing of international guidelines, through privacy-preserving analyses of real-world data at scale.

## Methods

This study was conducted in accordance with all relevant ethical regulations, with approval and data access permissions obtained from the HRA and the institutional review boards or equivalent ethics committees at each participating centre.

### Institutional data access & data protection approvals

The atomCAT2 study used FL methodology to develop prognostic models using local datasets from 16 participating centres (see Supplementary Material: Supplementary Note 2), without exchange of any sensitive individual patient-level data. Only non-identifiable aggregated information in the form of mathematical parameters, such as model coefficients, was shared between centres to train and validate the federated models. Each institution obtained approval from the relevant institutional review board or equivalent ethics committee for accessing and using patient data for research, and provided a copy of their local data access approval, including approving body and reference number, to the central study coordinator. For UK centres, central approval was provided by the Health Research Authority (HRA) (IRAS project ID: 303103, REC reference: 22/WA/0081). Full details of the approving body, approval reference, and informed consent status for each centre are provided in Supplementary Note 2.

The atomCAT2 study involved no protocol-specific intervention beyond standard-of-care radiotherapy. Given the retrospective, non-interventional nature of the study and the use of routinely collected, de-identified data that remained at the originating institutions, the requirement for individual patient informed consent was waived or deemed not required by the approving bodies, as detailed in Supplementary Note 2.

### Prospective study protocol & statistical analysis plan

A prospective study protocol and comprehensive statistical analysis plan were developed collaboratively, published[11], and registered in Open Science Foundation[48]. Details of data item selection and definition, outcome definition, model specification, sample size calculation, and model evaluation are available in the protocol[11]. Model development, validation procedure and results were reported in accordance with the TRIPOD statement and checklist[49] (see Supplementary Material: Supplementary Note 1). Figure 4 illustrates the workflow for the entire study, including previously published preliminary and preparatory work[12].

### Study design & patient population

Patients treated with radical intent external beam radiotherapy for primary anal squamous cell carcinoma, with or without concomitant chemotherapy, were included. Inclusion was restricted to patients treated with conformal radiotherapy techniques (3D conformal radiotherapy (3D-CRT) or IMRT/VMAT). Individuals treated with palliative intent, and patients who had received prior pelvic radiotherapy or brachytherapy (either primary or as boost treatment) were excluded. Patients were treated according to each participating centre's protocols, which consisted of concurrent radiotherapy and chemotherapy with varying regimens, or radiotherapy only.

### Outcome definitions

Three outcomes were employed: OS, LRC, and FFDM. The Core Outcome Research Measures in Anal Cancer (CORMAC) initiative has identified these as key outcome measures for anal cancer research[50]. Complete definitions for all three outcomes can be found in the study protocol[11], in brief: OS included death from any event, with censoring at last clinical follow-up; LRC events included lack of complete response at evaluation following chemoradiotherapy and locoregional disease progression during follow-up. Patients were censored at death, at last clinical follow-up, if undergoing abdominoperineal resection for non-disease related reasons, or in case of distant metastases; distant metastasis was defined as disease recurrence outside of the primarily treated volumes, including any non-irradiated nodes outside the pelvis or other organs e.g. lung or liver. Patients were censored at local recurrence, at death, or at last clinical follow-up. All times were calculated from first radiotherapy treatment date.

### Identification of relevant prognostic factors

To identify potential prognostic factors for the three outcomes, a systematic review of the literature was conducted[24]; this included studies published after 2000, with medium-to-large cohorts (>100 patients), reporting on anal cancer outcomes after treatment with conformal radiotherapy. Factors identified as prognostic through multivariable analysis in several studies were selected and prioritised (based on the number of studies reporting on them), which formed an initial list of relevant data to be collected. This list was reviewed by three senior clinical oncologists (AG, MGG, MB), who added additional relevant factors. A data scoping exercise across participating centres limited the list to factors consistently available in routine clinical data. HIV and HPV status, though clinically relevant, were excluded due to inconsistent availability and high risk of informative missingness across participating centres. Finally, a prioritised list of factors for each prognostic model (see Sample Size section below) was created by the senior oncologist team. A full description of the variable selection and correlation assessment process, including clinical rationale and literature sources, is provided in the published protocol[11]. A data dictionary was created and shared between all centres for standardised data collection, formatting, and reporting. Additional detail on missing data handling and standardisation procedures across centres is provided in Supplementary Material: Supplementary Note 3.

### Patient data collection & missing data

Patient data were identified and extracted from existing research and clinical databases at each participating site. To ensure good data quality, each institution spot checked all extracted data, ensuring adherence to the coding system specified in the data dictionary and identifying any outliers. The study protocol[11] pre-specified a framework on how to deal with missing data at individual centres, in multiple different scenarios, including use of data imputation. Only gross tumour volume (GTV, cm³) required use of imputation in two centre datasets; this was implemented based on a global median of means from all other centres.

### Sample size

A prospective sample size calculation was performed using the framework devised by Riley et al.[39], and implemented using the "pmsampsize" package in R, to determine the minimum sample size required to fit a Cox proportional hazards model for each of the three outcomes. The number of prognostic factors included in the final models was based on the total number of patients available in the primary consortium cohort; with the specific set of factors based on the prioritised list developed from a systematic literature review and expert oncologist input. The detailed methodology used to calculate sample size and prioritised list of factors is provided in the study protocol[11].

### Descriptive data analysis

Descriptive data analysis included calculation of summary and survival statistics from each centre. Overall summary statistics for continuous variables were calculated using weighted means. Two-year, three-year, and five-year OS, LRC, and FFDM were estimated by Kaplan-Meier methods. Potential follow-up times were based on the inverse Kaplan-Meier estimator[51].

### Model development & specification

Primary analysis consisted of development and internal validation (TRIPOD Type 2b[52]) of federated multivariable Cox Proportional Hazards models[31] across 14 centres; separately for OS, LRC, and FFDM. Secondarily, external validation (TRIPOD Type 3) was performed using previously unseen data from two additional institutions (see below). Based on available patient numbers in the 14 primary cohort centres ($n = 1428$), eight parameters could be included in the models.

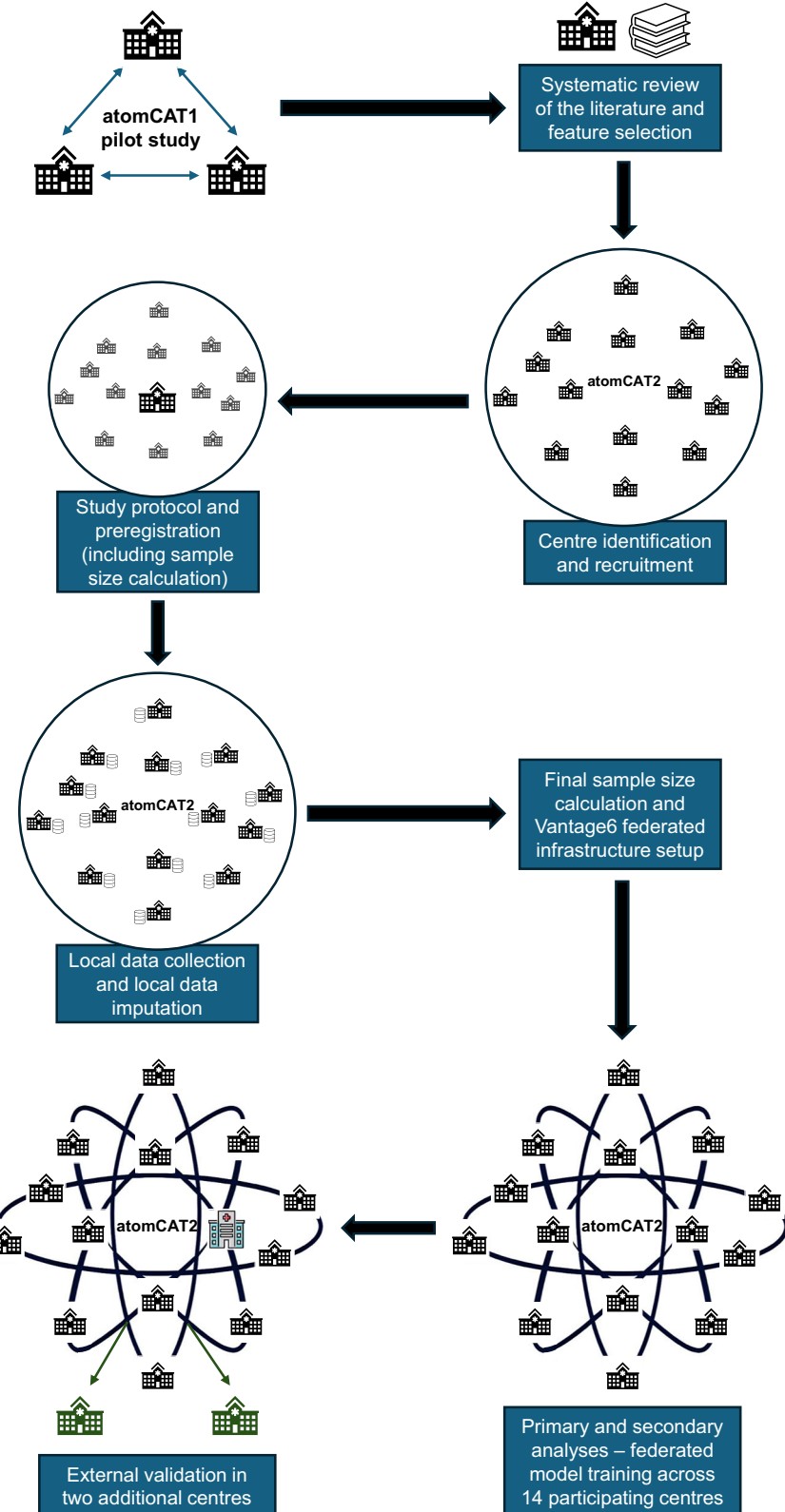

**Fig. 4 | atomCAT2 study design workflow.** Workflow of the atomCAT2 study, showing progression from a pilot network of three centres (atomCAT1) to the establishment of the 16 centre atomCAT network for federated model development and external validation.

Predictors used were: age, biological sex, T stage, nodal involvement, primary GTV (using a $\log_{10}$ transformation), prescribed primary tumour dose (in $EQD2_{\alpha/\beta=10Gy}$), and histology (as per the prioritisation lists in the study protocol). Biological sex was included as a predictor in all multivariable models and was recorded based on clinical

documentation at each participating centre. Gender identity was not separately collected or verified. Sex-specific analyses beyond inclusion as a covariate were not pre-specified and were not performed.

A set of secondary models were additionally developed to explore the impact of different factor parametrisation (e.g. for age and GTV),

different definitions (e.g. for tumour staging), and including secondary factors only available for a subset of patients (e.g. performance status). All analyses were run as multivariable models, with no data driven feature selection or model reduction. Patients with metastatic disease at baseline were excluded from all models for FFDM. An additional exploratory comparison between the federated models and a single-centre model (Centre 2) was performed to illustrate the added value of the federated approach.

## Federated model learning

Description of the FL architecture, as well as further technical details of the model training algorithm's aggregation process, convergence thresholds, and iteration scheme are provided in Supplementary Material: Supplementary Note 4. In brief, models were trained iteratively through exchange of local model coefficients and fit errors between each participating centre and a central server. At each iteration, the central server aggregated model coefficients computed locally in each centre and a single globally-convergent model was determined by minimising the total error[53–55]. The process was repeated until pre-specified convergence criteria were fulfilled. The source code for infrastructure implementation and the federated Cox regression algorithm utilised for all prognostic models are freely accessible on GitHub (see Supplementary Material: Supplementary Note 4).

## Model reporting

For each model developed, the estimated factor effects were reported as Hazard Ratios (HR), along with 95% confidence intervals (CI). Factors were deemed prognostic if their 95% CI did not overlap with 1. For each of the three outcomes, reference outcome rates at 2 years, 3 years and 5 years were calculated. The reference outcome rate can be defined as the outcome rate when all model factors are set to their baseline value. To calculate reference outcome rates, all categorical factors were set to 0, age at the start of radiotherapy was set to 35 years, prescribed dose to the primary tumour was set to 40 $EQD2_{\alpha/\beta=10Gy}$, and $\log_{10}$ of GTV was set to 0.02572 (GTV of 1.1 cm$^3$). The combination of reference outcome rates and factor effect estimates (HR) enables prediction of outcomes for future patients, rendering the models useable for individual patient outcome prediction.

## Evaluation and visualisation of model performance

Model performance was assessed using Harrell's concordance index (c-index, providing a measure of model discrimination)[56] on a per-centre basis, and a global weighted average calculated. A closed-loop internal-external "leave-one-centre-out" cross-validation method[57] was applied to obtain the out-of-sample performance for the primary study cohort, providing an initial estimate of the over-optimism of the global model. The weighted mean of c-index values generated in all centres was reported. For model calibration, predicted three-year OS, LRC, and FFDM rates for each patient were calculated based on the primary models, and patients were allocated to 'high risk' or 'low risk' groups according to the global predicted mean for each outcome. For each centre, the actual (observed) three-year outcome rates for 'high risk' and 'low risk' patients were evaluated using Kaplan-Meier estimates and subsequently plotted against the mean predicted outcome rates for each group to indicate model calibration. Finally, summary plots were generated to illustrate the observed weighted mean outcome rates for predicted 'high risk' and 'low risk' groups across centres at two, three, and five years.

## External validation

The models developed in the primary consortium cohort were further evaluated for their calibration and discrimination ability in two external validation cohorts. The external validation centres were selected based on cohort size, completeness of routinely collected variables, and geographical representation (one European and one Australian centre). Both were independent from the 14 training centres to ensure external validation across distinct populations. This process aimed to assess the reproducibility and generalisability of the models. Patient data collection and formatting followed the same study protocol as for the primary consortium cohort, and analyses retained the principles of no individual patient-level data sharing.

## Role of the funding source

None of the study sponsors have had any involvement in the study design; in the collection, analysis and interpretation of data; in the writing of the report; or in the decision to submit the article for publication.

## Reporting summary

Further information on research design is available in the Nature Portfolio Reporting Summary linked to this article.

## Data availability

Due to the nature of this research, the individual-level patient data analysed in this study cannot be made publicly available. These data include demographic variables, tumour characteristics, treatment details, and clinical outcomes collected at the participating centres. Even after de-identification, such data are considered potentially re-identifiable and fall outside the scope of the ethical approvals obtained for this study, which do not permit redistribution of patient-level data. The study protocol and statistical analysis plan, including a detailed data dictionary defining all variables, have been previously published (Theophanous et al., Diagn Progn Res 2022; https://doi.org/10.1186/s41512-022-00128-8). Aggregated results supporting the findings of this study are provided in the figures and tables of the main manuscript and Supplementary Materials. Source data underlying the reported figures are provided with this paper. Source data are provided with this paper.

## Code availability

The custom code developed for this study is publicly available at https://github.com/MaastrichtU-CDS/atomcat2 and is released under the GNU General Public License v3.0. The repository contains all scripts used for data processing, federated model execution, and aggregation of results. The code has been archived on Zenodo and assigned a persistent DOI[58]. The open-source Vantage6 (v2.3.4) federated learning infrastructure was used to carry out all federated learning analyses: https://github.com/vantage6/vantage6. Two Vantage6 federated learning algorithms were employed to run the federated analysis in this study. Federated Cox regression algorithm: https://github.com/IKNL/vtg.coxph, and federated validation algorithm: https://github.com/MaastrichtU-CDS/vtg.coxph_val. Descriptive data analysis was conducted in each participating centre individually using R/RStudio, and aggregation of results (including federated results) was carried out by the coordinating centre using R/RStudio.

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

## Acknowledgements

We would like to acknowledge Cancer Research UK funding (Grant Ref. C19942/A28832 and RRCOER-Jun24/100004) for the Leeds Radiotherapy Research Centre of Excellence (RadNet), which facilitated this work and provided academic salary support for Andrew Scarsbrook. Ane Appelt is supported by Yorkshire Cancer Research Academic Fellowship funding (grant L389AA). Leonard Wee is in receipt of funding from the Dutch Research Council (NWO) – BIONIC (grant no. 629.002.204), TRAIN (grant no. 629.002.212), CARRIER (grant no. 628.011.212), and STRaTegy (grant no. 14930), and from the Hanarth Foundation. Andrew Scarsbrook also acknowledges academic salary support from Leeds Hospitals Charity (Ref: 9R01/1403). None of the funders have had any involvement in the study design; in the collection, analysis and interpretation of data; in the writing of the report; or in the decision to submit the article for publication.

## Author contributions

A.A. and L.W. conceived the study idea. A.A., L.W., E.M., S.T., and P.I.L. established the atomCAT consortium and formulated the overarching research goals and aims. S.T. and A.A. coordinated the planning and execution of all research activity. A.A. and S.T. contributed to obtaining the required legal and ethics (HRA and REC) study approvals in the U.K. A.A., L.W., E.M., S.T., P.I.L., M.G.G., M.B., and A.G. designed the methodology. A.C. and L.W. contributed to the development of the federated learning infrastructure. A.C., L.W., S.T., and P.I.L. contributed to the technical implementation of federated learning. S.T., A.A., A.G., M.G.G., P.I.L., M.B., C.D., R.R., R.J., L.R., M.A.G., E.S., V.V., A.A.M., O.P., R.M., D.M., R.K., and M.F. were responsible for the local data collection at each participating centre. Members of the atomCAT consortium contributed to the local aspects of atomCAT2 in individual centres, including data collection, local dataset preparation, infrastructure setup, and local coordination. Please see the atomCAT Consortium section for the full list of atomCAT consortium members who contributed to this work. A.G., M.G.G., M.B., A.S., and D.S.M. contributed to the discussion of the clinical relevance of the study findings. S.T. and A.A. prepared the initial draft manuscript. L.W., E.M., P.I.L., A.G., M.B., M.G.G., A.D., A.S., and D.S.M. critically reviewed the draft manuscript and provided feedback. S.T. and A.A. revised the draft manuscript. All co-authors critically reviewed the updated version of the manuscript and provided feedback. S.T. and A.A. prepared the final version. All authors read and approved the final version of the manuscript.

## Competing interests

Leonard Wee receives consultancy fees when providing continuing professional development courses for radiotherapy physicists via Elekta AB (Stockholm, Sweden). Andre Dekker is a founder and employee of Medical Data Works B.V., which provides commercial support for Vantage6-based federated learning infrastructures. The remaining authors declare no competing interests.

## Additional information

[1]Leeds Teaching Hospitals NHS Trust, Leeds, UK. [2]Leeds Institute of Medical Research at St James's, University of Leeds, Leeds, UK. [3]Department of Medical Physics, Oslo University Hospital, Oslo, Norway. [4]Department of Radiation Oncology (Maastro), GROW Research Institute for Oncology and Reproduction, Maastricht University Medical Centre +, Maastricht, The Netherlands. [5]Department of Radiation Oncology, The Netherlands Cancer Institute - Antoni van Leeuwenhoek (NKI-AVL), Amsterdam, The Netherlands. [6]Department of Radiation Oncology, Amsterdam University Medical Center, Amsterdam, The Netherlands. [7]South Western Sydney Clinical Campus, School of Clinical Medicine, UNSW, Sydney, New South Wales, Australia. [8]Ingham Institute for Applied Medical Research, Liverpool, New South Wales, Australia. [9]Department of Radiation Oncology, Liverpool and Macarthur Cancer Therapy Centres, Sydney, New South Wales, Australia. [10]School of Electrical, Computer and Telecommunications Engineering, University of Wollongong, Wollongong, New South Wales, Australia. [11]Fondazione Policlinico Universitario A.Gemelli IRCCS, Università Cattolica S.Cuore, Rome, Italy. [12]Department of Oncology, Oslo University Hospital, Oslo, Norway. [13]Institute of Clinical Medicine, University of Oslo, Oslo, Norway. [14]Cambridge University Hospital NHS Foundation Trust, Cambridge, UK. [15]The Christie NHS Foundation Trust, Manchester, UK. [16]Goethe University Frankfurt, University Hospital, Department of Radiotherapy and Oncology, Frankfurt, Germany. [17]German Cancer Research Center (DKFZ), Heidelberg, Germany, German Cancer Consortium (DKTK), Partner Site, Frankfurt, Germany. [18]Frankfurt Cancer Institute (FCI), Frankfurt, Germany. [19]RWTH Aachen University Medical Centre, Aachen, Germany. [20]Center for Integrated Oncology Aachen Bonn Köln Düsseldorf (CIO ABCD), Aachen, Germany. [21]Department of Oncology, Oxford University Hospitals NHS Foundation Trust, Oxford, UK. [22]Department of Oncology, University of Oxford, Oxford, UK. [23]Champalimaud Foundation, Lisbon, Portugal. [24]Greater Poland Cancer Centre, Poznań, Poland. [25]Hull University Teaching Hospitals NHS Trust, Hull, UK. [26]School of Engineering, Cardiff University, Cardiff, UK. [27]Bank of Cyprus Oncology Centre, Nicosia, Cyprus. [28]Department of Radiation Biology, Institute for Cancer Research, Oslo University Hospital, Oslo, Norway. [34]These authors jointly supervised this work: Eirik Malinen, Leonard Wee, Ane Appelt. ✉e-mail: stelios.theophanous@nhs.net

## atomCAT Consortium

Richard Adams[26], Krystyna Adamska[24], Muhammad Amin[29], Ane Appelt[1,2,34], Maaike Berbee[4], Nikola Dino Capocchiano[11], Philip Chlap[7,8,9], Ananya Choudhury[4], Peter Colley[25], Andrea Damiani[11], Viola De Luca[11], Charlotte Deijen[5,6], Andre Dekker[4], Antri Demetriou[27], Shrikant Deshpande[7,9], Michael J. Eble[19], Anthony Espinoza[7,9], Emmanouil Fokas[16,17,18,30], Matthew Field[7,8,9,10], Maria Antonietta Gambacorta[11], Loukia Georgiou[27], Alexandra Gilbert[1,2], Marianne Grønlie Guren[12,13], Ann Henry[2], Andrew Hoole[14], Lois C. Holloway[8,9,10,31,32], Rashmi Jadon[14], Thomas Jansen[19], Tomas Janssen[5], Rohit Kochhar[15], Alexandros Kritikopoulos[27], Joanna Y. G. Lau[14], Mark T. Lee[7,9], John Lilley[1], Gisela Lima[25], Per-Ivar Lønne[3], Patricia Lopes[23], Adam Loveday[14], Eirik Malinen[28,34], Stefania Manfrida[11], Jenny Marsden[33], Daniel Martin[16,17,18], Carlotta Masciocchi[11], Joseph Mercer[15], Ahmed Allam Mohamed[19,20], Rebecca Muirhead[21,22], Elisavet Papageorgiou[27], Oriol Parés[23], Gareth Price[15], Thomas Rackley[29], Łukasz Raszewski[24], Claus Michael Rödel[16,17,18], Rajarshi Roy[25], Mariachiara Savino[11], Andrew Scarsbrook[1,2], Athina Sdrolia[25], David Sebag-Montefiore[2], Emiliano Spezi[26], Joep Stroom[23], Ioannis Stylianou[27], Stelios Theophanous[1,2]✉, David Thwaites[2,32], Maciej Trojanowski[24], Vincenzo Valentini[11], Rens van Haveren[5], Baukelien van Triest[5], Vassilios Vassiliou[27], Amy Walker[7,8,9,31] & Leonard Wee[4,34]

[29]Velindre University NHS Trust, Cardiff, UK. [30]University Hospital Cologne, Cologne, Germany. [31]Centre for Medical Radiation Physics, University of Wollongong, Wollongong, New South Wales, Australia. [32]Institute of Medical Physics, The University of Sydney, Sydney, New South Wales, Australia. [33]University Hospitals of Leicester NHS Trust, Leicester, UK.

