## [Peer Review File · Nature Communications]

An international multi-centre study to develop and validate federated learning-based prognostic models for anal cancer

Corresponding Author: Dr Stelios Theophanous

Version 0:

Reviewer comments:

Reviewer #1

(Remarks to the Author)

The study provides much-needed data on a rare cancer type, addressing a critical gap in oncology research and setting a benchmark for similar studies. It explores the use of Federated Learning (FL) to develop and validate prognostic models for anal cancer outcomes, leveraging real-world data from multiple international centers. The primary objective is to address the challenges of data scarcity and privacy concerns in rare cancer research by utilizing FL, which allows collaborative model training without sharing individual patient data. With data from 16 centers across Europe and Australia, this study represents one of the largest international cohorts of anal cancer patients. The geographic diversity adds to the generalizability of its findings.

The study included baseline patient and treatment characteristics data, and the development of federated multivariable Cox Proportional Hazards models for overall survival (OS), locoregional control (LRC), and freedom from distant metastases (FFDM). The models demonstrated good performance and calibration during cross-validation and external validation, with weighted c-indices of 0.68, 0.71, and 0.69 for OS, LRC, and FFDM, respectively.

The prospective planning, including sample size calculation and study pre-registration, highlights the rigor and reliability of the research design. The adherence to the TRIPOD guidelines further underscores the study's commitment to transparency and reproducibility. However there are some points needs to be highlighted :

The federated learning architecture had limitations in implementing standard prognostic model development steps like feature selection, advanced methods for missing data imputation, and correlation analysis.

The introduction/background effectively sets the stage but might benefit from a more compelling hook—perhaps a striking statistic or anecdote to draw the reader in. Similarly, a workflow to introduce the methodology could have presented a better demonstration about the whole study.

The models were validated both internally (using cross-validation) and externally (via independent datasets), ensuring the results are robust and less likely to be overfitted to the training data. The study addresses a critical gap, setting a benchmark for similar studies and the article discusses limitations, it could explicitly suggest how future studies might address these gaps. For instance, could emerging technologies improve federated learning methodologies.

The identification of prognostic factors like tumor stage, nodal involvement, and tumor volume provides actionable insights for clinical decision-making. These findings lend weight to the conclusion that FL can uncover critical prognostic information to inform personalized treatment strategies.

The study's conclusion touches on potential applications, but this could be further elaborated. Offering more examples of how the findings could influence clinical practice or inform policy might strengthen the impact.

Tables and figures are used well, but their clarity might be enhanced. For example, ensuring all figures have detailed captions and are self-explanatory could aid comprehension

There are many specialized terms and acronyms (e.g., OS, LRC, FFDM). Adding a glossary intended for wider dissemination could make the article more reader-friendly.

(Remarks on code availability)

N/A

Reviewer #2

(Remarks to the Author)

Overall, the manuscript presents an innovative federated learning approach to develop prognostic models based on a large, international, multi-center cohort of anal cancer patients. This approach holds promise for enhancing risk stratification in rare cancers by overcoming data privacy challenges and integrating heterogeneous data. However, several issues merit attention to strengthen the manuscript:

Major Issues:

1. The authors only presented the survival curves of all patients, but failed to compare the survival curves between the high - risk and low - risk groups predicted by the model. This is likely the most crucial result in a prognosis - related article, as it directly demonstrates the discriminatory power of the prognostic model for patient survival.
2. There is a lack of comparative analysis between the federated learning approach and simpler, single-center trained model or traditional clinical stratification methods (e.g., TNM stage), which limits the demonstration of the method's added value.
3. Table 3 only shows the results of univariate analysis, while lacking multivariate analysis to adjust for confounding factors. Multivariate analysis is essential to identify the independent prognostic factors and strengthen the reliability of the research findings.

Minor Issues:

1. The federated learning modeling in this study is conducted for a simple statistical model like Cox regression. It is unclear whether this training mode would still be applicable if other machine learning models or deep learning models were used. The authors are expected to discuss the potential challenges and feasibility when applying the federated learning approach to different types of models.
2. Some methodological details, such as the handling of missing data and standardization procedures across centers, are briefly described and would benefit from further elaboration.

(Remarks on code availability)

Reviewer #3

(Remarks to the Author)

I appreciate the opportunity to review the manuscript entitled "Federated Learning with Real-World Data: An International Multi-Centre Study to Develop and Validate Prognostic Models for Anal Cancer."

The authors present an innovative approach to developing prognostic models for rare cancers through federated learning, which enables collaborative analysis without sharing patient-level data—only model parameters are exchanged between centers. This strategy effectively addresses data privacy concerns and regulatory challenges.

From a clinical standpoint, this study is noteworthy. Its strengths include a robust multicenter design and an unusually large cohort for such a rare malignancy.

I would like to raise the following questions for the authors:

1. While the federated learning approach offers clear advantages regarding data protection and decentralization, could the authors elaborate on whether results would differ significantly if a traditional centralized model had been used? If available, a brief comparison or discussion on this point would enhance the manuscript.
2. Was HIV status included in the dataset? Although I understand that HPV status may often be unavailable, HIV infection is highly relevant in the context of anal cancer and is commonly recorded in clinical settings. If not included, please comment on this limitation.
3. Was there any available clinical information regarding prior anal condyloma disease? While indirect, this could serve as a surrogate marker for HPV infection and is usually easier to extract from medical records.
4. In your view, could these prognostic models influence or change current clinical decision-making for patients with anal cancer? If so, how might they be integrated into routine practice?

Once again, I thank you for the opportunity to review this important and well-executed study.

(Remarks on code availability)

Version 1:

Reviewer comments:

Reviewer #1

(Remarks to the Author)

Authors have addressed all the concerns, and the article can be accepted in its current form.

(Remarks on code availability)

Reviewer #2

(Remarks to the Author)

Thank you for your detailed clarifications. All of my previous concerns have now been satisfactorily addressed.

(Remarks on code availability)

Reviewer #4

(Remarks to the Author)

Major Comments

Federated Learning Methodology: Explanation of FL is minimal. Readers unfamiliar with the concept may not understand how it works.

Suggestion: Include a concise description of FL, aggregation algorithm (e.g., FedAvg, FedProx), number of iterations, convergence thresholds, error minimization process, handling of site-level variability, and correlation diagnostics or variable selection criteria for model stability.

External Validation Centers: Selection criteria for the two external centers are unclear.

Suggestion: Clarify how these centers were chosen to ensure representativeness and independence from the primary cohort.

Cohort Differences and Statistical Assessment: Clinically meaningful differences exist between primary and validation cohorts (sex, T stage, tumor volume, chemotherapy regimens).

Suggestion: Include formal statistical tests (chi-square, Fisher's exact, t-test, Mann-Whitney) to assess baseline differences and discuss potential impact on model performance and generalizability.

Clinical Relevance of Discrimination Metrics: C-indices are modest (OS: 0.68–0.72, FFDM: 0.69–0.79).

Suggestion: Discuss clinical implications and limitations of these discrimination metrics.

Single-Center Comparison

Federated learning is compared to a single-center model, but statistical support for improved generalizability is lacking. Suggestion: Include confidence intervals or statistical testing to strengthen this claim.

Variable Selection and Model Parameters: Rationale for including/excluding variables (performance status, treatment compliance, GTV, staging) is unclear, particularly regarding trade-offs with sample size.

Suggestion: Clarify decision process and limitations.

Radiotherapy Dose: Claim that prescribed dose is not prognostic contradicts prior literature.

Suggestion: Discuss possible confounding, cohort heterogeneity, or data limitations.

Clinical Novelty: Most identified prognostic factors (T stage, N stage, sex, GTV) are already known.

Suggestion: Clarify added clinical value beyond demonstrating FL methodology.

Minor Comments

Kaplan-Meier Analyses: Aggregated survival curves do not stratify by key clinical variables.

Suggestion: Consider subgroup KM curves (T/N stage, treatment regimen).

Follow-Up Data: Centres 6 and 14 excluded due to missing 5-year follow-up.

Suggestion: Acknowledge as potential source of bias for long-term predictions.

Treatment Completion and Response Assessment: Proportion completing planned RT/chemotherapy, timing, criteria for response, and classification of relapses are not reported.

Suggestion: Include these details to contextualize outcomes.

Calibration Interpretation: OS and FFDM calibration slightly worse than LRC; implications not discussed. Clarify how imperfect calibration affects clinical use.

Subgroup Analysis: Model performance in subgroups (elderly, high-risk T/N stage) not reported.

Suggestion: Include subgroup performance or discuss limitations for clinical decision-making.

Comparison with Centralized Models

Mathematical equivalence mentioned, but no empirical comparison. Clarify limitations of this assumption for real-world performance.

Missing Variables: HIV, HPV, and performance status excluded due to missing data; impact on confounding and generalizability is unclear. Discuss how this may bias results.

Speculative Future Applications:

Discussion of AI, LLMs, multimodal data is speculative. Clearly separate forward-looking statements from confirmed study results. Provide concrete guidance or scenarios for clinical application.

Formatting Consistency: The manuscript alternates between "centre" and "center." Standardize spelling throughout.

(Remarks on code availability)

Dear Reviewer 1,

Thank you for the reply and your thorough review of the manuscript. We very much appreciate your input and help on improving the manuscript. We have addressed your comments below.

The study provides much-needed data on a rare cancer type, addressing a critical gap in oncology research and setting a benchmark for similar studies. It explores the use of Federated Learning (FL) to develop and validate prognostic models for anal cancer outcomes, leveraging real-world data from multiple international centers. The primary objective is to address the challenges of data scarcity and privacy concerns in rare cancer research by utilizing FL, which allows collaborative model training without sharing individual patient data. With data from 16 centers across Europe and Australia, this study represents one of the largest international cohorts of anal cancer patients. The geographic diversity adds to the generalizability of its findings.

The study included baseline patient and treatment characteristics data, and the development of federated multivariable Cox Proportional Hazards models for overall survival (OS), locoregional control (LRC), and freedom from distant metastases (FFDM). The models demonstrated good performance and calibration during cross-validation and external validation, with weighted c-indices of 0.68, 0.71, and 0.69 for OS, LRC, and FFDM, respectively.

The prospective planning, including sample size calculation and study pre-registration, highlights the rigor and reliability of the research design. The adherence to the TRIPOD guidelines further underscores the study's commitment to transparency and reproducibility. However there are some points needs to be highlighted:

1. The federated learning architecture had limitations in implementing standard prognostic model development steps like feature selection, advanced methods for missing data imputation, and correlation analysis.

We thank the reviewer for their comment on the limitations of the federated learning infrastructure in accommodating standard model development steps, including feature selection, advanced imputation methods, and correlation analysis.

a. Feature selection:

We recognise that automated feature selection is a widely used step in model development workflows. However, recent research has underscored several challenges associated with automated approaches, such as LASSO and stepwise selection, including instability, inflated performance estimates, and a lack of generalisability to new datasets. Consequently, we opted against automated model-based feature selection in this study.

Instead, we employed a clinically guided feature selection process: a panel of anal cancer experts reviewed available clinical variables and reached consensus on relevant prognostic factors based on clinical experience and further informed by a systematic review of the literature. This approach aimed to promote clinical interpretability and to ensure that prognostic factors reflect current standards of care. Also, please note that our registered prospective protocol did not foresee automated feature selection. While automated feature selection techniques may be appropriate in future exploratory studies - for example, when evaluating large panels of candidate biomarkers - our aim here was to demonstrate the feasibility of a robust, reproducible model grounded in existing clinical knowledge.

We have now explicitly stated this as a strength in the **Discussion section (paragraph 4)** of the revised manuscript, as detailed below:

“Firstly, feature selection was based on expert consensus and literature review, rather than automated methods, avoiding known limitations with techniques like LASSO and stepwise selection in smaller, heterogeneous datasets⁴⁴⁻⁴⁶. This approach ensured clinical relevance, reproducibility, and supported prospective sample size calculation.”

The “limitations” section in the **Discussion section (paragraph 6)** was also amended to reflect this. The lack of automated feature selection in this study is no longer mentioned as a limitation in the following sentence:

“This includes advanced methods for missing data imputation⁵⁰⁻⁵¹ and correlation analysis.”

b. Missing data imputation:

The current version of the Vantage6 infrastructure does not support advanced methods for federated imputation. In this study, we used centre-specific mean or mode imputation methods, as these were the only viable options that preserved patient data privacy. We fully acknowledge this limitation in the manuscript and consider it an important direction for future methodological development of the federated learning platform.

c. Correlation analysis:

Correlation analysis between predictors was inherently part of the feature selection process. Variables known or expected to exhibit high collinearity were either excluded or explored in secondary models. For example, we examined the relationship between T-stage and gross tumour volume (GTV), and found that although the two are correlated, they capture overlapping yet non-identical prognostic information.

These two aspects are now clarified in the **Discussion section (paragraph 6)** of the revised manuscript:

“Instead, simple imputation methods were used, and correlated variables were handled during the study design phase. For example, secondary models explored the overlap between T stage and GTV size and indicated that some (but not all) of the information from GTV size can be captured by T stage.”

2. The introduction/background effectively sets the stage but might benefit from a more compelling hook—perhaps a striking statistic or anecdote to draw the reader in. Similarly, a workflow to introduce the methodology could have presented a better demonstration about the whole study.

In response to this constructive feedback, we have added a new sentence in the introduction that highlights the rising incidence of anal cancer in Europe over recent decades, in order to better engage the reader and underscore the clinical relevance of our work. We also added a brief example to emphasise the challenge of treating under-represented subgroups of patients, such as older patients, where prospective clinical trial data are particularly limited. This example is revisited in the Discussion to contextualise the relevance of our findings.

The following section is included in the **Introduction section (paragraph 3)** of the revised version of the manuscript:

“However, its incidence is rising substantially, with studies showing a 35% increase in men and a 75% increase in women between 1988 and 2012 across Europe¹⁵. This trend is largely attributed to increased exposure to high-risk HPV, following demographic and behavioural shifts since the introduction of the oral contraceptive pill in the 1960s. Despite the rising incidence, most centres treat relatively few cases per year and may struggle to identify appropriate evidence-based treatment for patient subgroups, many of whom are under-represented in clinical trials. As an example, if an oncologist were to see an elderly patient with

anal cancer in their clinic today, the treatment offered might differ substantially across cancer centres and countries¹⁶.”

To contextualise and further highlight the relevance of our findings, the following section was added to the Discussion section (paragraph 3) of the revised manuscript:

“Importantly, older patients comprised nearly one-third of our cohort, yet remain poorly studied in prospective trials¹⁶. Models trained on RWD may therefore offer a tool to support individualised treatment discussions and improve prognostication for such under-represented groups.”

In addition, a workflow diagram outlining the methodological approach has been added in the **Methods section (Figure 1)**, to provide a clearer overview of the study design and structure.

3. The models were validated both internally (using cross-validation) and externally (via independent datasets), ensuring the results are robust and less likely to be overfitted to the training data. The study addresses a critical gap, setting a benchmark for similar studies and the article discusses limitations, it could explicitly suggest how future studies might address these gaps. For instance, could emerging technologies improve federated learning methodologies.

We appreciate this constructive comment. We agree that it is important to highlight how future studies might address current limitations of federated learning methodologies. In the revised Discussion section, we have added examples of emerging technologies and approaches that could enhance federated learning in practice. These include the use of large language models (LLMs) and multimodal generative pre-trained transformer (GPT) systems to support data collection, curation, and harmonisation; adoption of common data models (CDMs) and interoperability frameworks (such as the OMOP CDM) to enable collaboration across heterogeneous datasets; the application of generative AI locally to create synthetic patient data for improved imputation or bias mitigation; and advances such as quantum cryptography and next-generation telecommunications to enable secure, low-latency infrastructures. These developments have the potential to further strengthen federated model performance while preserving privacy.

A new paragraph was added in the **Discussion section (paragraph 7)** to address this, which reads as follows:

“Future enhancements to federated learning are likely to come from advances in data curation, interoperability, and infrastructure. Large language models and multimodal generative pre-trained transformer (GPT) systems could assist with local data cleaning, mapping, and harmonisation, reducing the burden of manual curation. Adoption of common data models (CDMs) and interoperability frameworks, such as the OMOP CDM⁵² and the operational ontology for oncology (O3)⁵³, can enable collaboration and analysis across heterogeneous clinical datasets. Generative AI could also be deployed locally to create synthetic patient data to support imputation or mitigate inclusion bias without sharing identifiable information.”

4. The identification of prognostic factors like tumor stage, nodal involvement, and tumor volume provides actionable insights for clinical decision-making. These findings lend weight to the conclusion that FL can uncover critical prognostic information to inform personalized treatment strategies. The study’s conclusion touches on potential applications, but this could be further elaborated. Offering more examples of how the findings could influence clinical practice or inform policy might strengthen the impact.

We thank the reviewer for this valuable suggestion. We have now expanded the Conclusion (last paragraph of Discussion section) to emphasise how the study’s findings may inform clinical practice and policy. Specifically, we highlight the potential for using federated risk scores to support personalised treatment planning and follow-up, as well as for guiding evidence-based decision-making in policy through large-scale privacy-preserving analyses. We also reinforce the relevance of our approach in enabling biomarker-driven precision oncology, particularly for underrepresented groups of patients.

The **final paragraph of the Discussion section** in the revised manuscript was expanded as follows:

“Our robust and validated prognostic models, developed using large cohorts, lay the foundation for decision support tools that can inform shared decision-making, enable more personalised treatment strategies, and may be considered in prospective clinical trials for patient inclusion and stratification. In the future, this federated approach could also guide biomarker-driven subgroup analyses and support policy development, such as reimbursement decisions for new therapies and informing of international guidelines, through privacy-preserving analyses of real-world data at scale.”

5. Tables and figures are used well, but their clarity might be enhanced. For example, ensuring all figures have detailed captions and are self-explanatory could aid comprehension. There are many specialized terms and acronyms (e.g., OS, LRC, FFDM). Adding a glossary intended for wider dissemination could make the article more reader-friendly.

Thank you for this suggestion. In response, we have reviewed all table and figure captions to ensure they are fully self-contained and explanatory. The legends of **Table 1, 2, and 3** were updated accordingly.

We have also added a glossary table in **Supplementary Material: Appendix 6** to define key acronyms and specialised terms, in order to support wider accessibility and reader comprehension.

Dear Reviewer 2,

Thank you for the constructive and useful comments on this manuscript. We appreciate the review and believe that your suggestions have contributed positively to the manuscript and have improved the content significantly. Replies to individual comments are provided below.

Overall, the manuscript presents an innovative federated learning approach to develop prognostic models based on a large, international, multi-center cohort of anal cancer patients. This approach holds promise for enhancing risk stratification in rare cancers by overcoming data privacy challenges and integrating heterogeneous data. However, several issues merit attention to strengthen the manuscript:

Major Issues:

1. The authors only presented the survival curves of all patients, but failed to compare the survival curves between the high - risk and low - risk groups predicted by the model. This is likely the most crucial result in a prognosis - related article, as it directly demonstrates the discriminatory power of the prognostic model for patient survival.

The reviewer raises an important point regarding stratification of cancer and survival outcomes by predicted risk. We want to highlight that the discriminatory power of the model is already demonstrated in Figure 4 (previously Figure 3), which compares survival (as well as other cancer outcomes) between high- and low-risk groups at 3 years across all participating centres. Due to privacy constraints around sharing individual-level time-to-event data, as well as technical limitations of the federated infrastructure (Vantage6 did not support survival curve generation at the time of analysis), we were unable to produce individual-level Kaplan-Meier (KM) curves stratified by model-predicted risk across all centres. However, we have now addressed this by generating summary plots of observed weighted mean outcome rates across centres at two, three, and five years, for patients in predicted high- and low-risk groups. These have been added to the **Supplementary Material: Appendix 5**, and we now report the three-year high vs low-risk summary outcomes for each endpoint in the **Results section (paragraph 5)**, as follows:

“Observed weighted mean three-year outcome rates across centres were markedly different between predicted high- and low-risk groups: 73% vs 90% for OS; 76% vs 91% for LRC, and 83% vs 94% for FFDM. Plots for high- and low-risk groups, indicating the observed weighted mean outcome rates across

centres at two, three, and five years are presented in Supplementary Material: Appendix 5.”

2. There is a lack of comparative analysis between the federated learning approach and simpler, single-center trained model or traditional clinical stratification methods (e.g., TNM stage), which limits the demonstration of the method’s added value.

We appreciate the reviewer’s observation regarding the lack of direct comparison between the federated models and simpler approaches such as single-centre models. We agree this is an important point and have conducted a supplementary analysis using Cox regression models for OS, LRC, and FFDM, trained solely on data from a single-centre cohort, one of the largest in the study (Centre 2, n = 210). According to the sample size calculation, the single-centre models would only be able to support 2 parameters given the size of the cohort. However, to allow for a direct comparison of model performance to the federated models, the same parameters as the federated models were analysed. We then assessed the single-centre models’ performance on the two external validation cohorts to directly compare their generalisability with that of the federated models.

The resulting weighted mean C-indices across the two external validation cohorts for the primary overall survival (0.60), locoregional control (0.70), and freedom from distant metastases (0.62) models were substantially lower than those observed for the federated models (0.72, 0.75, and 0.79, respectively). Additionally, the estimated HRs were considerably different, in some cases (e.g. for the effect of concomitant chemotherapy HR=2132183.193) massively so. This highlights the reduced generalisability of single-centre models and underscores the robustness of the federated, multi-institutional approach. These results have now been added to the **Results section (paragraph 7)** of the revised manuscript:

“To illustrate the added value of the federated approach, comparator models for all three outcomes were developed using the same specifications, but only data from a single centre (Centre 2, n=210). When evaluated on the external validation cohorts, this single-centre models yielded substantially lower C-indices for all outcomes (OS: 0.60, LRC: 0.70, FFDM: 0.62) compared to the federated models (0.72, 0.75, and 0.79, respectively), indicating reduced generalisability.”

3. Table 3 only shows the results of univariate analysis, while lacking multivariate analysis to adjust for confounding factors. Multivariate analysis is essential to identify the independent prognostic factors and strengthen the reliability of the research findings.

We apologise for the lack of clarity in the original legend. Table 3 does in fact present the results from multivariable Cox regression models. To avoid any ambiguity, we have updated the table legend to explicitly state that the results are from multivariable analysis, as follows:

“Table 3. Results from multivariable Cox proportional hazards models for overall survival (OS), locoregional control (LRC), and freedom from distant metastases (FFDM), trained on the full primary cohort (14 centres). Hazard ratios (HRs) and 95% confidence intervals (CI) are shown for each included variable. All models were adjusted for the full set of covariates listed (apart from radiotherapy technique, which was not included in the FFDM model).”

Minor Issues:

1. The federated learning modeling in this study is conducted for a simple statistical model like Cox regression. It is unclear whether this training mode would still be applicable if other machine learning models or deep learning models were used. The authors are expected to discuss the potential challenges and feasibility when applying the federated learning approach to different types of models.

We thank the reviewer for raising this point. The Vantage6 platform is designed to be model-agnostic and can support a wide range of algorithms, including machine learning and deep learning approaches. While our study focused on Cox regression, the platform can accommodate other model types, provided they are containerised for federated execution. For example, recent implementations using Vantage6 include federated survival forests and convolutional neural networks for medical imaging analysis. We have added a brief note in the **Discussion section (paragraph 7)** with references to these developments:

“The Vantage6 platform supports containerised implementation of a large range of model types, including machine learning and deep learning algorithms. Recent examples include federated survival forests and convolutional neural networks for radiotherapy and imaging research⁵⁴, illustrating the potential for broader applications as the infrastructure matures further.”

2. Some methodological details, such as the handling of missing data and standardization procedures across centers, are briefly described and would benefit from further elaboration.

Thank you for this helpful suggestion. While detailed methodological procedures are provided in our published study protocol (Theophanous et al, Diagn Progn Res 2022), we agree that including a standalone, consolidated summary would improve clarity for readers. We have therefore added a new appendix (**Supplementary Material: Appendix 3**) that provides additional detail on missing data handling, local data standardisation procedures, and data quality assurance steps across centres. This is now referenced in the **Identification of Relevant Prognostic Factors subsection** of the Methods section, as follows:

“Additional detail on missing data handling and standardisation procedures across centres is provided in Supplementary Material: Appendix 3.”

Dear Reviewer 3,

The authors would like to thank you for the review and for taking the time to help us improve the manuscript. We have addressed your comments below.

I appreciate the opportunity to review the manuscript entitled "Federated Learning with Real-World Data: An International Multi-Centre Study to Develop and Validate Prognostic Models for Anal Cancer."

The authors present an innovative approach to developing prognostic models for rare cancers through federated learning, which enables collaborative analysis without sharing patient-level data—only model parameters are exchanged between centers. This strategy effectively addresses data privacy concerns and regulatory challenges.

From a clinical standpoint, this study is noteworthy. Its strengths include a robust multicenter design and an unusually large cohort for such a rare malignancy.

I would like to raise the following questions for the authors:

1. While the federated learning approach offers clear advantages regarding data protection and decentralization, could the authors elaborate on whether results would differ significantly if a traditional centralized model had been used? If available, a brief comparison or discussion on this point would enhance the manuscript.

We appreciate this insightful question. A centralised model could not be implemented in our setting, as our study was explicitly designed to address the challenges associated with data sharing across institutions. For this reason, patient-level data were never pooled, and a traditional centralised analysis was inherently prevented. However, prior work by Lu et al. (2015) has shown that the federated Cox model used in our study, under the Breslow approximation, is mathematically equivalent to the corresponding centralised model when the same data are used. This suggests that our federated implementation should yield results equivalent to a centralised approach, while maintaining patient data privacy.

To address this question, the following sentence was added to the **Discussion section (paragraph 3)**:

“Although comparison to a fully centralised model was not feasible due to data governance constraints, the federated Cox regression algorithm used in this

study has been shown to be mathematically equivalent to a centralised model under the Breslow approximation when applied to the same data³¹.”

2. Was HIV status included in the dataset? Although I understand that HPV status may often be unavailable, HIV infection is highly relevant in the context of anal cancer and is commonly recorded in clinical settings. If not included, please comment on this limitation.

Thank you for raising this important question. Prior to finalising the study protocol, we conducted a scoping exercise with participating centres and clinical collaborators to identify variables that were both clinically relevant and consistently available across institutions. HIV status was excluded due to high levels of missingness and concerns about informative missingness across centres. While routinely collected in the coordinating centre for new patients, availability and completeness varied significantly across participating institutions. This reflects broader trends: in the UK, for example, most HIV-positive patients are diagnosed early via screening and often present with early-stage disease, resulting in relatively low representation in retrospective treatment cohorts, as indicated by the short-term results from the PLATO-ACT4 trial (Gilbert et al., 2025).

We have added this rationale to the Methods section, which now reads as follows:

“A data scoping exercise across participating centres limited the list to factors consistently available in routine clinical data. HIV status, though clinically relevant, was excluded due to inconsistent availability and high risk of informative missingness across participating centres. Finally, a prioritised list of factors for each prognostic model (see Sample Size section below) was created by the senior oncologist team.”

3. Was there any available clinical information regarding prior anal condyloma disease? While indirect, this could serve as a surrogate marker for HPV infection and is usually easier to extract from medical records.

We acknowledge the reviewer’s interest in prior anal condyloma. However, this information is not routinely collected in clinical practice and is generally poorly recorded outside of high-risk groups. While anal condyloma is associated with HPV infection and increased risk of developing anal cancer, there is currently no evidence that its prior presence influences treatment outcomes. For this reason, it was not included in the analysis.

4. In your view, could these prognostic models influence or change current clinical decision-making for patients with anal cancer? If so, how might they be integrated into routine practice?

We thank the reviewer for this important point. As outlined in our response to Reviewer 1, Comment 4, we have expanded the Conclusion to better highlight how these models could support personalised treatment decisions and inform clinical practice and policy, particularly in underrepresented patient subgroups. We believe that this updated conclusion adequately addresses this comment as well.

Once again, I thank you for the opportunity to review this important and well-executed study.

Dear Reviewer #4,

Thank you for the detailed review of the manuscript and the constructive feedback. Below, we provide a point-by-point response to your comments. Where appropriate, we indicate proposed clarifications or edits included in the manuscript.

Major Comments

1. Federated Learning Methodology: Explanation of FL is minimal. Readers unfamiliar with the concept may not understand how it works.

Suggestion: Include a concise description of FL, aggregation algorithm (e.g., FedAvg, FedProx), number of iterations, convergence thresholds, error minimization process, handling of site-level variability, and correlation diagnostics or variable selection criteria for model stability.

We thank the reviewer for highlighting this point. The overall federated workflow is already described in **Methods - Federated Model Learning** and in **Supplementary Material: Supplementary Note 4**. However, we agree that a short summary for readers less familiar with federated learning (FL) would be helpful. We have therefore expanded Supplementary Note 4 to explicitly state the aggregation approach, number of iterations, and convergence thresholds:

“During federated model training, each participating centre independently fitted a local Cox model on its own data and transmitted only aggregated summary statistics - specifically, the first- and second-order partial derivatives of the log-likelihood (gradient and Hessian) - to the central coordinating server. No individual-level patient data were exchanged. The central aggregator then updated the global parameter vector by summing site-level contributions to the global gradient and Hessian, following the framework described by Lu et al. Model parameters were iteratively updated until convergence, typically achieved within 6-10 global aggregation rounds. Convergence was defined as a change in the global log-likelihood of less than 1×10^{-5} between consecutive iterations. This iterative optimisation ensured that the federated solution was mathematically equivalent to the pooled maximum-likelihood estimator under the Breslow approximation, while maintaining full data privacy.”

We have also added a brief explanatory sentence in the **Methods - Federated Model Learning** section referring readers to these details:

“Description of the FL architecture, as well as further technical details of the model training algorithm’s aggregation process, convergence thresholds, and iteration scheme are provided in Supplementary Material: Supplementary Note 4.”

The rationale and process for variable selection are already extensively described and justified in **Methods - Identification of Relevant Prognostic Factors** and in our published protocol (Theophanous et al., Diagn Progn Res 2022). We will add a cross-reference at the end of that subsection to ensure clarity:

“A full description of the variable selection and correlation assessment process, including clinical rationale and literature sources, is provided in the published protocol¹¹.”

2. External Validation Centers: Selection criteria for the two external centers are unclear.

Suggestion: Clarify how these centers were chosen to ensure representativeness and independence from the primary cohort.

We agree that additional transparency on the choice of validation centres will be useful. We have clarified in **Methods - External Validation** that these centres were selected based on a combination of pragmatism (based on available cohort size and data completeness, taking into account the rarity of this cancer type) and geographical representativeness (one European and one Australian centre). Their data were entirely independent from the 14 training centres. We have made this rationale explicit in the revised text:

“The models developed in the primary consortium cohort were further evaluated for their calibration and discrimination ability in two external validation cohorts. The external validation centres were selected based on cohort size, completeness of routinely collected variables, and geographical representation (one European and one Australian centre). Both were independent from the 14 training centres to ensure external validation across distinct populations.”

3. Cohort Differences and Statistical Assessment: Clinically meaningful differences exist between primary and validation cohorts (sex, T stage, tumour volume, chemotherapy regimens).

Suggestion: Include formal statistical tests (chi-square, Fisher’s exact, t-test, Mann–Whitney) to assess baseline differences and discuss potential impact on model performance and generalisability.

We appreciate this comment but note that our principal purpose was not to compare groups statistically, but to evaluate the model’s robustness across differing populations. Performing hypothesis tests between independent datasets would be methodologically inappropriate and could mislead interpretation.

The presence of heterogeneity between centres is essential for developing generalisable models, since prognostic models trained only on homogeneous data would have limited external validity. Meaningful prediction depends on learning from variation in demographics, staging, and treatment. In this context, inter-institutional differences are not a flaw but a necessary feature of the federated design, allowing the model to identify relationships that persist despite non-identically distributed data.

The validation phase explicitly tests performance across independent cohorts to ensure that the model generalises beyond the characteristics of the training data. We have clarified this rationale in the **Discussion section (paragraph 3)** with the following text:

“Formal statistical comparisons between the training and validation cohorts were not performed, as these were independent datasets used to test model generalisability across different populations. The observed differences in demographics and treatment reflect real-world variability, which the federated model is designed to accommodate.”

4. Clinical Relevance of Discrimination Metrics: C-indices are modest (OS 0.68–0.72, FFDM 0.69–0.79).

Suggestion: Discuss clinical implications and limitations of these discrimination metrics.

We thank the reviewer for this observation. While the C-indices are moderate, they are consistent with other externally validated prognostic models in oncology. We have added the following text in the **Discussion section (paragraph 3)**:

“While the discrimination metrics from the outcome models were moderate (c-indices 0.68 - 0.79), these values are comparable with other externally validated real-world prognostic models in oncology in general and anal cancer specifically^{28–30}. They indicate clinically meaningful performance in a rare cancer setting, where predictive model performance is often limited by cohort size and heterogeneity.”

5. Single-Centre Comparison: Federated learning is compared to a single-centre model, but statistical support for improved generalisability is lacking.

Suggestion: Include confidence intervals or statistical testing to strengthen this claim.

We thank the reviewer for this point. The single-centre comparison was performed as a post-hoc exploratory analysis, added at the suggestion of a reviewer during the previous round. It was not part of the original statistical analysis plan and was intended solely to illustrate the difference in behaviour between federated and single-site models, not to serve as a formal hypothesis test.

The purpose of this comparison was not to establish statistical superiority of federated over single-centre learning, but to demonstrate that single-centre model results lack generalisability. Prognostic models trained on data from a single centre often reflect the local patient population and treatment protocols, and thus perform less well when applied to more heterogeneous populations. In contrast, the federated approach enables model training across a large, diverse, and geographically distributed dataset, which is essential for the development of robust and generalisable models.

As this was an unplanned descriptive analysis, further statistical testing would not be methodologically justified.

6. Variable Selection and Model Parameters: Rationale for including/excluding variables (performance status, treatment compliance, GTV, staging) is unclear, particularly regarding trade-offs with sample size.

Suggestion: Clarify decision process and limitations.

The variable selection process is described in detail in **Methods - Identification of Relevant Prognostic Factors** and the published study protocol. Variables were chosen based on a systematic review of the literature, where factors consistently reported as prognostic across multiple cohorts were considered for inclusion and for the initial ranking. Additional expert clinical input was used to further rank and prioritise candidate factors. The final list of potential factors was further informed by data completeness across centres. Altogether, this approach avoided automated feature selection, incorporated prior published information as well as expert knowledge, and allowed for prospective sample size calculation.

7. Radiotherapy Dose: Claim that prescribed dose is not prognostic contradicts prior literature.

Suggestion: Discuss possible confounding, cohort heterogeneity, or data limitations.

We agree this point should be explicitly clarified. We have added a sentence in the **Discussion (paragraph 2)** stating that the absence of a significant dose effect may reflect confounding by indication and inter-centre heterogeneity, and that ongoing prospective studies such as PLATO are expected to provide more information on this issue:

“This lack of dose-effect relationship could reflect cohort heterogeneity and/or confounding by indication, as higher doses tend to be prescribed to patients with more advanced disease. In addition, variations in dose reporting across institutions may dilute this relationship. Prospective, randomised studies with risk-adapted doses such as PLATO²⁷ are expected to clarify this issue.”

We would additionally note that previously published dose-response models (such as Muirhead et al. [Radiother Oncol 2015]) are based on literature meta-analyses, and thus are just as likely to suffer from confounding by indication.

8. Clinical Novelty: Most identified prognostic factors (T stage, N stage, sex, GTV) are already known.

Suggestion: Clarify added clinical value beyond demonstrating FL methodology.

We appreciate the reviewer’s observation. We will emphasise that the study’s novelty lies in successfully demonstrating the feasibility and reproducibility of federated learning for rare cancer modelling, not in redefining known prognostic factors. We have rephrased the first paragraph of the **Discussion (paragraph 1)** to highlight this point more clearly, as follows:

“Although some of the prognostic variables identified (e.g., T stage, N stage, GTV) are well supported in previous studies, our findings show that federated learning can reproduce and externally validate these variables. This highlights the capability of federated learning for developing robust, privacy-preserving prognostic models for rare cancers using real-world data.”

Minor Comments

1. Kaplan–Meier Analyses: Aggregated survival curves do not stratify by key clinical variables.

Suggestion: Consider subgroup KM curves (T/N stage, treatment regimen).

We thank the reviewer for this suggestion. Due to federated data-privacy constraints, generating centre-level subgroup KM curves is not feasible within the current infrastructure. The high- versus low-risk plots (**Results - paragraph 5**) serve as an alternative visualisation of model discrimination. We have noted this limitation explicitly in the **Discussion (paragraph 7)**:

“Due to privacy restrictions within the federated learning framework, individual-level Kaplan-Meier curves stratified by clinical subgroups could not be generated, although future Vantage6 algorithms may allow secure aggregation of subgroup survival estimates.”

2. Follow-Up Data: Centres 6 and 14 excluded due to missing 5-year follow-up.

Suggestion: Acknowledge as potential source of bias for long-term predictions.

We agree with the point raised and we have added a short statement in the **Discussion (paragraph 7)** section acknowledging this as a possible limitation for longer-term calibration, although it is unlikely to affect the three-year outcomes that were the study’s predefined endpoints.

“Moreover, two centres were excluded from the summary 5-year outcome rates due to incomplete follow-up; however, this represents a small fraction of the total cohort (14 centres), and the two- and three-year endpoints remain unaffected.”

3. Treatment Completion and Response Assessment: Proportion completing planned RT/chemotherapy, timing, criteria for response, and classification of relapses are not reported.

Suggestion: Include these details to contextualise outcomes.

We thank the reviewer for this comment. The proportion of patients completing planned radiotherapy and chemotherapy was deliberately not included in the current analysis,

as the (predefined) main study aim was to evaluate *pre-treatment* factors prognostic for long term outcomes, to potentially facilitate treatment individualisation.

Response evaluation is notoriously tricky in anal cancer [Glynne-Jones et al. (Lancet Oncol 2017; 18: 347-356)], and not straightforward to consistently define across all participating centres. Consequently, we deliberately did not set out to predict treatment response, but rather longer-term cancer control.

Full details of all endpoint definitions, including the definition and classification of relapses can be found in the published study protocol. We believe that referring readers directly to the protocol provides sufficient context and avoids unnecessary repetition. Accordingly, we do not plan to modify the manuscript text based on this comment, but we have ensured that the published protocol is clearly cited as the source for these methodological details.

4. Calibration Interpretation: OS and FFDM calibration slightly worse than LRC; implications not discussed.

Suggestion: Clarify how imperfect calibration affects clinical use.

We have added a concise explanation in the **Discussion (paragraph 3)**, noting that modestly lower calibration for OS and FFDM likely reflects the broader heterogeneity in non-cancer mortality across centres. Calibration for these outcomes remained within acceptable limits, supporting the model's practical interpretation.

“Moreover, slightly poorer calibration for OS and FFDM relative to LRC was observed, which likely reflects greater heterogeneity in non-cancer mortality across centres. Despite this, model performance remained within acceptable limits for clinical interpretation.”

5. Subgroup Analysis: Model performance in subgroups (elderly, high-risk T/N stage) not reported.

Suggestion: Include subgroup performance or discuss limitations for clinical decision-making.

We acknowledge this as a valid consideration. Subgroup analyses were not part of the prespecified statistical analysis plan. We have noted this explicitly in the **Discussion section (paragraph 7)**, and mentioned that such analyses will be feasible in future federated studies as larger datasets become available. Keeping tumour and node stage separate, rather than as a high-risk group, enable the outcomes of these factors to be studied separately, although this could be interesting to look at after the publication of PLATO.

“Subgroup analyses were not prespecified and were underpowered in the available cohorts. Future federated learning studies with larger datasets will allow more reliable evaluation of subgroups, including older, frailer, and comorbid patients.”

6. Comparison with Centralised Models

Mathematical equivalence mentioned, but no empirical comparison. Clarify limitations of this assumption for real-world performance.

We have expanded the **Discussion (paragraph 7)** to clarify that while the federated Cox model is mathematically equivalent to its centralised counterpart under the Breslow approximation, small numerical deviations may arise from asynchronous updates or local preprocessing. These differences are expected to be negligible in practice.

“Additionally, although the federated Cox model is mathematically equivalent to the centralised version under the Breslow approximation, small numerical deviations may occur due to asynchronous updates or local preprocessing. These differences are likely to be negligible and do not affect overall model conclusions.”

7. Missing Variables: HIV, HPV, and performance status excluded due to missing data; impact on confounding and generalisability is unclear.

Suggestion: Discuss how this may bias results.

We thank the reviewer for highlighting this point. We have added a brief discussion noting that HIV and HPV status were excluded due to inconsistent availability across centres. HPV status is not yet used to modify standard treatment, and HIV-positive patients with well-controlled disease generally receive the same therapy as HIV-negative patients. In the **Discussion section (paragraph 6)**, we have also referenced that frailty, rather than age, is emerging as a clinically important factor warranting investigation in future federated studies:

“HIV and HPV status were excluded due to inconsistent data availability across centres, and performance status was omitted because of high missingness and potential informative bias. Frailty may ultimately represent a more clinically meaningful predictor of treatment tolerance and outcome than performance status or age; however, its incorporation into federated analyses would require standardised definitions to ensure comparability across centres.”

8. Speculative Future Applications: Discussion of AI, LLMs, multimodal data is speculative.

Suggestion: Clearly separate forward-looking statements from confirmed study results; provide concrete guidance for clinical application.

The section discussing potential future applications of AI, large language models, and multimodal data integration was added at the request of a reviewer during the earlier review round. Since these additions have already been reviewed and approved by the reviewer, we prefer not to modify this section further to maintain consistency with the previously accepted revisions.

The statements are already framed as forward-looking perspectives and do not alter the interpretation of the study's core findings. Therefore, we do not plan any changes to the manuscript in response to this comment.

9. Formatting Consistency: The manuscript alternates between "centre" and "center." Standardise spelling throughout.

We have ensured that British English ("centre") is used consistently throughout the text, but have maintained alternative spelling where relevant (e.g. "center" when part of an institution name).